# Response of Rice with Overlapping Growth Stages to Water Stress by Assimilates Accumulation and Transport and Starch Synthesis of Superior and Inferior Grains

**DOI:** 10.3390/ijms231911157

**Published:** 2022-09-22

**Authors:** Xinpeng Wang, Jinxu Fu, Zhaosen Min, Detang Zou, Hualong Liu, Jingguo Wang, Hongliang Zheng, Yan Jia, Luomiao Yang, Wei Xin, Bin Sun, Hongwei Zhao

**Affiliations:** Key Laboratory of Germplasm Enhancement, Physiology and Ecology of Food Crops in Cold Region, Ministry of Education, Northeast Agricultural University, Harbin 150030, China

**Keywords:** overlapping growth stages, drought stress, yield formation, starch synthesis, superior and inferior grains

## Abstract

Drought stress at jointing–booting directly affects plant growth and productivity in rice. Limited by natural factors, the jointing and booting stages of short-growth-period rice varieties are highly overlapped in high-latitude areas, which are more sensitive to water deficit. However, little is known about the dry matter translocation in rice and the strategies of starch synthesis and filling of superior and inferior grains under different drought stress was unclear. In this study, the rice plants were subjected to three degrees of drought stress (−10 kPa, −25 kPa, −40 kPa) for 15 days during the jointing–booting stage; we investigated dry matter accumulation and translocation, grain filling and enzyme activities to starch synthesis of superior and inferior grains in rice with overlapping growth stages from 2016 to 2017. The results showed that drought stress significantly reduced dry matter accumulation in the stems and leaves. Mild and moderate drought increased dry matter translocation efficiency. However, severe drought stress largely limited the dry matter accumulation and translocation. A large amount of dry matter remains in vegetative organs under severe drought stress. The high content in NSC in stem and sheath plays a key role in resisting drought stress. The drought stress at jointing–booting directly caused a change in the grain filling strategy. Under moderate and severe drought, the grain-filling active period of the superior grains was shortened to complete the necessary reproductive growth. The grain-filling active period of the inferior grains was significantly prolonged to avoid a decrease in grain yield. The significant decrease in the grain-filling rate of the superior and inferior grains caused a reduction in the thousand-grain weight. In particular, the influence of the grain-filling rate of inferior grains on the thousand-grain weight was more significant. Drought stress changed the starch synthesis strategies of the superior and inferior grains. Soluble starch synthase and starch branching enzyme activities of inferior grains increased significantly under drought stress. GBSS activity was not sensitive to drought stress. Therefore, amylose content was decreased and amylopectin synthesis was enhanced under drought stress, especially in inferior grains.

## 1. Introduction

Drought has become the main cause of crop yield reduction worldwide. For example, China suffered an agricultural drought in 2016; the affected area was 9872.7 thousand hectares, according to the National Bureau of Statistics (NBSC) [1]. Because the reduction in crop yield caused by drought causes enormous economic disruption, demand for the development of drought-tolerant crops is increasing [2]. Locally adapted drought-stress tolerance traits are required to achieve maximal crop yield potential [3].

In 2020, the rice (*Oryza sativa* L.)-cultivated area achieved 30,080 thousand hectares in China, with 211.9 million tons (NBSC) [4] of yield. As one of the world’s most important cereals, rice is also the most water-intensive crop. Compared with other cereal crops, rice cultivars have been planted with zero soil water potential for thousands of years, which causes greater sensitivity to decreases in soil-water potential. Rice production requires large amounts of water resources. Three to five thousand liters of water was used for 1 kg of rice seed production, while other crops, such as maize or wheat, required less than half of that [5]. Therefore, understanding the range of physiological changes initiated by drought stress is important for developing supportive measures to enhance drought resistance.

Grain production is mainly sourced from non-structural carbohydrates (NSCs) stored in vegetative organs before anthesis and produced after anthesis [6]. In the process of starch accumulation in grains, drought decreases carbon metabolism, causing carbon starvation and even death in severe cases [7]. According to McDowell’s research, drought stress leads to a lack of water in cells, which results in poor material exchange. Drought stress also influences carbohydrate transportation and the use of carbohydrates for metabolism and defense, which intensifies its impact on carbon metabolism and results in severe carbon starvation [7].

Many studies have found that plant growth stagnation precedes the reduction in photosynthesis during drought stress, which leads to a carbohydrate surplus [8,9,10,11,12]. McDowell’s work suggests that the result of carbohydrate surplus is that the decrease in growth requirements was more significant than the reduction in photosynthesis under drought stress. The reduction in photosynthesis was greater than the consumption of maintaining respiration. Carbohydrate content in the whole plant increases at the beginning of drought but eventually declines if drought stress reduces photosynthesis significantly for an extended period. Carbohydrates, including respiratory metabolism and osmotic regulation, are still used to maintain cell survival [7]. The transport rate and volume of carbohydrates stored in vegetative organs before anthesis can affect the grain-filling rate and serve as an essential base material for initiating grain filling. Some research has shown that the initiation of grain filling is related to the transport of carbohydrates stored in the stem and sheath to spikelets at the heading stage. When carbohydrates stored in the stem and sheath are below a certain threshold, they cannot be transferred to grains in large quantities, even though photosynthate accumulation is high after anthesis, thus, affecting grain plumpness [13].

Mild alternate wetting–drying at the filling stage was beneficial for transporting NSC in the stem and sheath to grains [14,15,16]. The results showed that the transport of photosynthetic assimilates from the source to the sink played an essential role in grain filling. Grain filling in rice panicles directly determines rice quality and yield and is closely related to the grain position in the panicle [17]. The superior grains grew on the primary branch at the top of the panicle anthesis. In contrast, the inferior grains germinated on the secondary branch at the bottom of panicle anthesis. When compared with superior grains, inferior grains usually do not have a better grain-filling rate and thousand-grain weight [18,19]. These results indicate that the superior and inferior grains might have different starch synthesis strategies.

Grain filling in inferior grains is believed to require a large amount of substance and energy accumulation necessary for carbohydrate metabolism. The activities of invertase, starch synthase and ADPG-PPase (AGPase) in superior grains are significantly higher than those in inferior grains in the early and middle time of the filling stage [20,21,22,23,24]; water deficit can improve the critical enzyme activity in starch synthesis in inferior grain by increasing grain filling rate under −25 kPa at the filling stage [25]. This confirmed that the crucial enzyme activity of starch synthesis in inferior grains lower than superior grains is harmful to grain filling [24]. Severe drought stress during the filling stage could promote carbon transfer from stem to grain, which increased the grain-filling rate and the plumpness of inferior grains but significantly reduced grain yield. At the same time, researchers found that the time to maximum grain-filling rate (Tmax) of the inferior grain reached earlier under drought stress. The filling period shortened, sucrose and soluble sugar contents in inferior grains increased significantly and the starch accumulation rate was slow. This suggests that low soil water content promotes the grain-filling process in superior grains. However, the transforming ability of starch in inferior grains is reduced under drought stress. This asynchronism of the grain-filling process and uneven distribution of assimilates in the superior and inferior grains caused chalkiness [26]. Drought stress at the jointing–booting stage could reduce the grain number per panicle and grain plumpness and, thus, significantly reduce thousand-grain weight and yield [27]; the reduction range is 30.40–53.06% [28]. Drought stress from panicle primordium differentiation to the heading stage resulted in the degradation of spikelets, a decrease in seed setting rate and grain plumpness and, thus, significantly reduced yield [29]. However, some scholars believe that mild drought during the panicle primordium differentiation stage or reduction–division stage could alleviate the degradation of spikelets, thus, increasing the seed setting rate of rice [30,31].

Rice in cold regions has a short growth period. The jointing and booting stages of these varieties are highly overlapped and vegetative growth and reproductive growth occur simultaneously. There is also a high overlap between the heading and anthesis stages. The overlapping characteristics of growth stages also determine its extreme sensitivity to water during the transition period of the growth stage, especially at the jointing–booting stage. In the high-latitude area, the frost-free period is short and the rice filling is characterized by a high filling intensity and short filling time. This characteristic indicates that the difference in grain filling between the superior and inferior grains was larger than that in other areas. The kind of damage that water deficiency at the jointing–booting stage will cause grain filling and yield formation in high-latitude areas and the effects of different degrees of drought stress on starch synthesis in the superior and inferior grains need to be investigated.

In this study, we selected two typical rice varieties in cold regions with overlapping growth stages. We set three different degrees of drought stress treatments at the jointing–booting stage (mild, moderate and severe drought stress), to explore the differences in matter accumulation, transport and grain filling between superior and inferior grains under different water-deficit conditions.

## 2. Results

### 2.1. Changes in NSC Content Response to Drought Stress in Stems and Sheaths

It was found that NSC content in the stems and sheaths increased significantly after 12 days of mild drought stress at the jointing–booting stage (Figure 1). The results of the two years showed that the NSC content in the stem and sheath of the two varieties had different changes under severe drought stress. The content in NSC in the stem and sheath of SJ6 (Figure 1a,c) under severe drought stress was even higher than that under moderate drought stress and there was no significant difference between SJ6 and the control. The NSC content in DN425 (Figure 1b,d) under severe drought stress was not significantly different from that under moderate drought stress. After restoration irrigation, the NSC content in DN425 was substantially higher than that under moderate drought stress. Notably, NSC remobilization of SJ6 occurred after D6 days under drought stress. The NSC remobilization initiation point of DN425 was significantly earlier than that of SJ6 and the content in NSC increased significantly after the onset of water stress. Drought stress at the jointing–booting stage also significantly affected the NSC content in the leaves (Appendix A).

### 2.2. Differences in Dry Matter Accumulation in Stems and Leaves under Different Drought Stress

Drought stress significantly reduced the dry matter accumulation in stems (stems and sheaths) and leaves during the stress period and the maximum dry matter accumulation also significantly decreased (Figure 2). Severe drought stress significantly reduced the dry matter accumulation rate of stems and sheaths (Figure 2a,b,e,f). Mild and moderated drought stress delayed the time to reach the maximum dry matter in stems and sheaths in 2017, but no such phenomenon was found in 2016, which needs further research. The results of two years showed that the dry matter accumulation of stems and sheaths under severe drought stress decreased slowly after reaching the maximum value, significantly lower than that of other treatments, and even surpassed the control level at the maturity stage.

The results of two years of the study showed that a short period (6 days) of mild drought stress could promote the dry matter accumulation of leaves to some extent (Figure 2c,g). The dry matter of leaves decreased in advance under mild and moderate drought stress. Severe drought stress immediately led to a decrease in leaf dry matter accumulation. However, after the decrease to a certain extent, the accumulation of dry matter began to rise, even though it was still under drought stress. Similar to the dry matter accumulation of stems and sheaths, the dry matter accumulation of leaves decreased slowly after reaching the maximum value under severe drought stress. A large amount of dry matter was retained in the vegetative organs, resulting in a great limitation of dry matter transport after anthesis under severe drought (Table 1). Mild and moderate drought stress significantly increased the dry matter translocation efficiency in stems and leaves after anthesis. It is noteworthy that dry matter in stems and leaves continued to increase after heading stage under severe drought stress, indicating that severe drought stress changed the inherent metabolic strategy of rice with overlapping growth stages.

### 2.3. Difference in Grain Filling between Superior and Inferior Grains under Drought Stress

Drought stress significantly decreased the grain-filling rate of rice in the cold region and there were differences between the two varieties under different degrees of drought stress (Table 2). The results showed that the GRmean of SJ6 under moderate and severe drought stress was significantly higher than that under mild drought stress. The GRmean of DN425 under moderate drought stress was the lowest, which was significantly lower than that under mild and severe drought stress. Under mild and moderate drought stress, the GRmax of the superior grains was significantly lower than that of the control. However, under severe drought stress, the GRmax of superior grains was not significantly different from that of the control. The GRmean and GRmax of the inferior grains of the two varieties decreased with soil water potential. The results of two years showed that the Tmax of the superior grains of DN425 was significantly shortened under mild drought stress and significantly delayed under severe drought stress. Under severe drought stress, the Tmax of DN425 was significantly shortened, but not in SJ6. Compared with the control, the active period of the superior grains (D) of grain filling increased significantly under mild and moderate drought stress and decreased significantly under severe drought stress. Drought stress significantly prolonged the active period of inferior grains of grain filling and the more severe the drought, the more pronounced the effect.

### 2.4. Effects of Drought Stress on Panicle Traits of Rice with Overlapping Growth Stages

Table 3 lists the panicle traits of the two cultivars over two years. Compared with the control, drought stress significantly reduced the secondary branch number and grain number of secondary branches of rice in the cold region. The more severe the drought, the greater the number of secondary branches. In the two-year experiment, the average number of secondary branches in SJ6 treatments decreased by 23.44%, 36.58% and 53.16%, respectively, compared with the control. The grain number of the secondary branches decreased by 21.06%, 34.61% and 54.83%, respectively. Compared with the control, the number of secondary branches in the DN425 treatments decreased by 22.63%, 36.35% and 49.71%, respectively. The grain number of secondary branches decreased by 17.75%, 35.50% and 56.09%, respectively, with significant differences among all treatments. The above results showed that panicle length and number of secondary branches decreased significantly under drought stress. The decrease in the number of secondary branches led to a significant reduction in the number of secondary branches. The more severe the drought, the more they reduce.

### 2.5. Effects of Drought Stress on Yield Components in Rice

As shown in Table 4, the number of spikelets per panicle was significantly reduced under drought stress. However, the difference in spikelets per panicle under mild and moderate drought stress was not significant, significantly higher than that under severe drought stress. The spikelets per panicle of SJ6 treatments (two-year average) were 14.51%, 19.70% and 32.68% lower than the control. The values of DN425 treatments were 11.90%, 19.45% and 31.62% lower than that of the control, respectively. Drought stress decreased the thousand-grain weight of rice and the more severe the drought, the more significant the reduction in thousand-grain weight. Compared with the control, the two-year results showed that the thousand-grain weight of SJ6 treatments decreased by 2.97%, 4.06% and 6.36% and the theoretical yield decreased by 25.14%, 27.67% and 43.70%, respectively. The thousand-grain weight of DN425 treatments decreased by 4.72%, 5.78% and 8.53%, respectively, and the theoretical yield decreased by 23.35%, 32.92% and 45.37%, respectively.

### 2.6. Difference between Starch Composition of Superior and Inferior Grains

Rice in cold region is popular for its high-quality taste, which is mainly due to its low amylose and high amylopectin ratio. To further explore the effect of drought stress on starch composition, we determined the amylose and amylopectin contents in the superior and inferior grains at the maturity stage. Drought stress had a significant effect on the amylose content in the inferior grains during the maturity stage (Figure 3a,c). The results showed that the amylose content in inferior grains of SJ6 (two-year average) under moderate and severe drought stress was significantly lower than that of the control, which decreased by 10.03% and 15.61%, respectively. The amylose content in the DN425 treatments decreased by 13.04% and 12.37%, respectively.

Drought stress significantly affected amylopectin content in the superior and inferior grains (Figure 3b,d). The amylopectin content in the superior grains under moderate and severe drought stress was significantly lower than that of the control. The results of two years showed that SJ6 under moderate and severe drought stress was 2.37% and 5.50% lower than that of the control, respectively. DN425 was 4.27% and 7.06% lower than that of the control, respectively.

Drought stress significantly increased the amylopectin content in the inferior grains. The more severe the drought, the higher the amylopectin content. The amylopectin content in the inferior grains of SJ6 treatments (two-year average) was 1.81%, 4.88% and 7.21% higher than that of the control, respectively. Moderate and severe drought stress treatments of DN425 were 2.02% and 4.06% higher than that of the control and the mild drought stress treatment showed interannual differences.

### 2.7. Physiological Differences in Starch Synthesis between Superior and Inferior Grains

#### 2.7.1. Soluble Starch Synthase

There was no significant difference in the soluble starch synthase (SSS) activity of superior grains among SJ6 treatments (Figure 4a). The SSS activity of DN425 severe drought treatment was significantly lower than that of other treatments from R5 to R15 days and reached a maximum at R20 days (Figure 4b). The maximum SSS activity of moderate and severe drought treatments was significantly lower than that of the control, decreasing by 20.69% and 26.74%, respectively, and there was no significant difference between the two treatments.

SSS activity of the SJ6 inferior grain (Figure 4c) increased slowly during drought stress. After irrigation, the SSS activity of SJ6 under moderate and severe drought conditions was significantly higher than that of the control after R5 days. The maximum SSS activity increased by 21.95% and 39.69% compared to that of the control, respectively. The SSS activity of all DN425 treatments reached a maximum at R20 (Figure 4d). Compared with the control, it was significantly higher in the moderate and severe drought treatments by 27.93% and 39.09%, respectively. There was no significant difference between the mild drought treatment and control groups. The SSS activity of DN425 treated with severe drought was significantly higher than that of the control at R25.

#### 2.7.2. Granule-Bound Starch Synthetase

GBSS activity in the superior and inferior grains did not change significantly under drought stress, except for severe drought stress (Appendix A). GBSS activity was significantly lower under severe drought stress from D9 to D12 days. This may be because panicle development was affected by severe drought stress and delayed the filling process.

#### 2.7.3. Starch Branching Enzyme

The SBE activity of SJ6 superior grains in severe drought treatment was significantly lower than that of the control and the SBE activity on D15 days decreased by 54.69% (Figure 5a). After irrigation, the SBE activity of SJ6 severe drought treatment was lower than that of the control and maintained an upward trend. The increased rate of SBE activity in the DN425 mild drought treatment was the fastest and reached the maximum at R15 days, which was significantly higher than that of other treatments and increased by 25.34% compared with the control (Figure 5b). The SBE activity of the moderate and severe drought treatments was consistently lower than that of the control after R15 days.

During the drought stress period, the SBE activity of the inferior grains of the two varieties in all treatments did not change (Figure 5c,d). After irrigation, the maximum SBE activity of SJ6 three drought stress treatments (A1, A2, A3) was significantly higher than that of the control, which increased by 42.89%, 45.95% and 30.32%, respectively (Figure 5c). The SBE activity of DN425 drought stress treatments was higher than that of the control during grain filling, but the difference was not significant (Figure 5d).

### 2.8. Correlation between Dry Matter Accumulation, Transportation and Grain-Filling

It can be seen from the table that the yield has a significant positive correlation (*p* < 0.01) with dry matter translocation (Table 5), efficient panicle number and spikelets per panicle and a significant positive correlation with grain-filling rate (IGRmean) in inferior grains. In addition, there was a significant positive correlation (*p* < 0.01) between the thousand-grain weight and the IGRmean of the inferior grains.

In conclusion, the efficient panicle number, spikelets per panicle and thousand-grain weight determine the yield. Dry matter translocation mainly affects yield by affecting the spikelets per panicle. The grain-filling rate of the superior and inferior grains indirectly affects the yield, mainly by affecting the thousand-grain weight. In particular, the inferior grain-filling rate had the most significant effect on the thousand-grain weight.

## 3. Discussion

### 3.1. Different Degrees of Drought Stress May Change the Rule of Dry Matter Accumulation

Many studies have shown that drought stress can promote carbohydrate transport [13,32]. Some scholars have found that soil drought after heading causes several NSCs in the rice stem to transfer to the grain. The contribution rate could even reach more than 40% [33] and some research found that this transport promotion effect would increase with an increase in drought stress intensity [14]. We found that the NSC content in stems and sheathing of SJ6 under severe drought was higher than that of the moderate treatment during the stress period, while that of DN425 occurred after the restoration of irrigation. Although the tested varieties showed different variation patterns, they all indicated that plants need more NSCs to resist the adverse effects of drought under severe drought stress, even considering the concentration of NSC content caused by the decrease in dry matter. According to the variation scale of NSC content in leaves and stems, NSC content in stems plays a pivotal role in plant resistance to drought stress. Compared with SJ6, the content in NSC in stems and sheaths of DN425 responded more quickly to drought stress, which may have positive significance for drought resistance. Interesting was the phenomenon that the peak of NSC contents were not the same timing with the peak of leaf dry weights under mild and moderate drought stress. When the content in NSC reached the peak, dry matter accumulation of leaves had begun to decrease due to drought stress. Drought stress leads to a decrease in dry matter, which inevitably leads to a decrease in photosynthetic capacity of leaves (Appendix A). Thus, reducing photosynthate synthesis further exacerbates the effects of drought stress. Therefore, a large number of NSCs are transported to leaves to maintain cell osmotic pressure to resist drought stress. This hypothesis was indirectly supported by the significant decrease in NSC content in stem sheath during moderate drought stress. The above phenomenon did not occur in severe drought, which may be because the photosynthetic rate of leaves under severe drought stress did not increase first and then decrease as it did under mild and moderate drought stress, but continued to decrease from the beginning of drought.

Dry matter accumulation is the basis of rice yield formation. It has been found that mild soil drought can promote the transport of dry matter to panicles [28]. The results showed that drought stress significantly decreased dry matter accumulation in stems and leaves. Under mild and moderate drought stress, dry matter transfer in rice leaves was advanced. Under severe drought stress, dry matter accumulation in rice leaves in the cold region was almost stopped. Under mild and moderate drought stress, dry matter accumulation in leaves decreased in advance. This may be because dry matter accumulation in the stem and sheath was significantly reduced due to drought stress and dry matter in leaves was forced to be transported in advance to maintain the growth of other organs. Mild and moderate drought stress promoted dry matter transport in the stems, sheaths and leaves after anthesis. Although mild and moderate drought stress promoted material transport and the net photosynthetic rate of leaves did not decrease significantly after restoring irrigation compared with the control (Appendix A), it did not achieve higher grain weight. The possible reason is that mild and moderate drought stress significantly reduced dry matter accumulation in stems and leaves and the limited material base could not maintain the original “Sink” (grain-filling) demand. Our earlier study found that a large number of spikelets degenerated after heading led to a decrease in grain number per panicle, which also confirmed this conjecture [34]. Dry matter accumulation of severe drought stress after anthesis was not reduced and even increased. A possible reason is that stem sheath dry matter accumulation has fallen dramatically during drought stress. Dry matter accumulation in leaves stagnate and the requirements for plant growth and grain filling cannot be satisfied. Therefore, the recovery of organs after drought stress requires a large amount of material after irrigation, which also leads to the retention of a large amount of dry matter in the vegetative organs after the resumption of irrigation. Failure to transport large quantities of materials to grains over time is bound to result in reduced yields. Dry matter translocation was reduced significantly after anthesis, which also confirmed the above conclusion. The cause of abnormal dry matter translocation efficiency of the stem, sheath and leaves of DN425 in 2017 is still unclear and requires further study. Under severe drought stress, the dry matter transshipment of stems and sheaths of both varieties decreased significantly, indicating that severe drought stress could not promote dry matter transport, instead of greatly restricting the process.

### 3.2. Difference in Grain-Filling Strategies between Superior and Inferior Grains under Drought Stress

Grain filling is the most crucial factor in determining rice yield, defining yield traits, such as grain weight, seed setting rate and quality [35]. Due to the difference in flowering time and nutrient competition between superior and inferior grains, the starting time of filling was different [36]. Some scholars have found that mild drought stress increases grain weight, maximum grain-filling rate and mean grain-filling rate of inferior grains, but has no significant effect on superior grains. Severe drought stress significantly decreased in the superior and inferior grains [14]. In this study, mild drought stress significantly reduced the grain-filling rate and significantly prolonged the grain-filling activity period of the superior and inferior grains in the two varieties. The results indicate that the active filling period of rice in the cold region was extended to compensate for the lack of filling rate under mild drought stress. It is worth noting that, compared with the mild drought stress treatment, the active filling period of the superior grains was significantly shortened and the inferior grains were significantly prolonged under moderate drought stress. This was in spite of a difference between the two varieties in grain-filling rate of superior grain under moderate drought stress. The moderate drought stress had a significant impact on the grain filling of the superior grain, so the filling time of the inferior grain was extended to make up for the deficiency in the filling state of the superior grain. Under severe drought stress, the Tmax of the superior grain of the two varieties was significantly prolonged and the Tmax of the inferior grain of DN425 was significantly shortened. Compared with the moderate drought stress, the active filling period of the superior grain was further shortened and the active filling period of the inferior grain was further prolonged. The results indicated that severe drought stress reduced the grain-filling rate of rice and significantly restricted the increase in the grain-filling rate of the superior grain. The active filling period of the superior grain was significantly shortened, so the filling time of the inferior grain should be extended to compensate for the effect of insufficient grain filling of superior grains on grain yield.

In conclusion, the grain-filling rate of rice was significantly reduced under drought stress at the jointing–booting stage and the effect on superior grains was more significant than that on inferior grains. The deficiency in grain-filling rate was mainly caused by prolonging the active filling period, which was consistent with the results of Yang et al. [37]. No significant correlation was observed between the average filling rate of the superior grains and yield. When the grain-filling rate further decreased with the soil water potential, the superior grain filling time was shortened to complete the necessary reproductive growth. The filling activity period of the inferior grains was extended significantly to avoid a dramatic reduction in yield.

### 3.3. Effects of Dry Matter Transportation on Yield Formation under Drought Stress

Some researchers have observed that drought stress decreases grain number per panicle and seed setting rate significantly at any time point from early panicle differentiation to booting stage [38]. Xu et al. reported that both mild and severe drought stress decreased the efficiency of panicle number. In contrast, mild drought increased grain number per panicle, thousand-grain weight and seed setting rate, while severe drought had the opposite effect [14]. In this study, panicle length and the number of secondary branches was significantly reduced under drought stress. The decrease in the number of secondary branches resulted in a decrease in the number of secondary branches (Appendix A). This indicated that drought stress mainly reduced the number of secondary branches and reduced the yield. The reduction in spikelet numbers, especially inferior spikelets, can help to maintain the full potential TGW under the source ability (viz. leaf photosynthesis and dry matter remobilization) was limited under drought stress. ANOVA also confirmed that drought stress had significant effects on grain number per panicle and thousand-grain weight of rice. Combined with the above, the decrease in grain number per spike was mainly caused by the significant reduction in the number of secondary branches under drought stress. The significant reduction in the thousand-grain weight indicated that although the grain-filling duration of inferior grains could be extended to compensate for the lack of grain filling under drought stress significantly, the effect was minimal. The correlation analysis further showed that the filling rate of the superior and inferior grains indirectly affected the thousand-grain weight to reduce the yield, especially the filling rate of the inferior grains.

### 3.4. Effects of Drought Stress on Starch Synthesis of Superior and Inferior Grains

The results showed no significant change in the SSS activity of the superior grain in SJ6 treatments compared to the control. However, the SSS activity of the inferior grain under moderate and severe drought stress was significantly higher than that of the control. The SSS activity of DN425 superior grains under severe drought stress was significantly lower than that of the control. The maximum SSS activity of DN425 under moderate drought stress was lower than that of the control. The maximum SSS activity of DN425 inferior grains under moderate and severe drought stress was significantly higher than that of the control. The results also showed that the starch synthesis ability of inferior grains was strengthened under moderate and severe drought stress. Some studies have reported a low effect on GBSS activity during the filling stage [39]. In this study, the GBSS activity of superior and inferior grains in the treatments was not significantly different, indicating that GBSS activity is not sensitive to drought stress. The GBSS activity of severe drought stress was significantly lower than that of the other treatments in the binging. This may be because panicle development is affected by severe drought stress and delayed the filling process. GBSS activity increased rapidly later with no significant difference between the other treatments, which also proved this point.

Xu et al. found that mild drought stress could enhance the SBE activity of inferior grains, while the SBE activity of both superior and inferior grains decreased under severe drought stress [14,40]. In this study, the SBE activity of SJ6 superior grains under severe drought stress was significantly lower than that of the control. Although the SBE activity of DN425 superior grains under mild drought stress was significantly lower than that of the control, it increased rapidly and was significantly higher than that of the control after the resumption of irrigation. SBE activity was consistently lower than that of the control under severe drought stress. The SBE activity of the inferior grains of the two varieties in all treatments was higher than that of the control in the filling stage, which was consistent with the results of previous research. These results indicate that drought stress promoted starch synthesis in inferior grains. The amylopectin synthesis of superior grains was affected under moderate and severe drought stress and the effect of severe drought stress was the most significant. In contrast, amylopectin synthesis of inferior grains was strengthened to compensate for the adverse effects of superior grains. The SSS and SBE activity of the inferior grains increased slowly at the beginning of the filling stage. This may be because the filling time of inferior grains is delayed compared with that of superior grains.

Higher amylopectin content was helpful to improve the taste of rice. In order to investigate the effect of changing starch synthesis strategy on rice quality under drought stress, amylose and amylopectin of superior and inferior grains at maturity were detected. Compared with the control, the amylose content in the two varieties of superior grains decreased slightly under drought stress. At the same time, the amylose content in inferior grains decreased significantly, which is consistent with the research results of Cai et al. [41]. Compared with the control, the amylopectin content in the superior grains decreased significantly under moderate and severe drought stress. The amylopectin content in the inferior grains was significantly higher than that of the control. The amylose and amylopectin contents in the superior grains decreased under drought stress, which may be due to the increased protein proportion in grains. Previous studies have confirmed this hypothesis [41]. The increase in amylopectin content in inferior grains is closely related to SBE activity under drought stress. These results indicated that drought stress increased the cooking quality of inferior grains and decreased the cooking quality of superior grains. The main reason was that drought stress changed the activities of SSS and SBE in superior and inferior grains. Under drought stress, the starch synthesis ability of inferior grains was enhanced, mainly in the proportion of amylopectin. Although the amylose content in inferior grains decreased under drought stress, it was not regulated by GBSS activity. We hypothesized that there was competition between amylose and amylopectin synthesis under drought stress. Although the activity of amylose synthetic-related enzymes was relatively stable under drought stress, drought stress induced an increase in SBE activity, which ultimately tilted the starch synthesis strategy towards amylopectin synthesis.

## 4. Material and Methods

### 4.1. Plant Material and Growth Conditions

The research was conducted in a rain-proof shelter of A’Cheng experimental site of the Northeast Agricultural University in Harbin City, China (126°40′ E, 45°10′ N), during the rice-growing season (April to September) in 2016 and 2017. The frost-free period is short in high latitudes and the suitable season for rice growth is only from May to October. The average daily temperature during the growing season at the experimental site is shown in Figure 6. In order to make the research results accurately reflect the response to drought stress, two overlapping rice cultivars with similar growth periods but different genotypes were selected as experimental materials, namely, grain number type and grain weight type. Two typical japonica rice varieties with overlapping growth periods were Songjing 6 (SJ6, grain number type, 135 days growth period and 2500 °C effective accumulated temperature) and Dongnong 425 (DN425, grain weight type, 140 days growth period and 2550 °C effective accumulated temperature). The average values for the selected soil characteristics of composite topsoil samples (0–20 cm) from the main experimental plots were as follows: organic matter was 20.34 ± 0.34 g kg^−1^; total N was 1.52 ± 0.09 g kg^−1^; total P was 0.49 ± 0.06 g kg^−1^; slowly available K was 654.5 ± 4.34 mg kg^−1^; available N was 1 mol L^−1^; NaOH-alkali-hydrolyzed N was 129.8 ± 4.34 mg kg^−1^; available P was 0.5 mol L^−1^; NaHCO_3_-Olsen P was 16.3 ± 0.5 mg kg^−1^; available K was 1 mol L^−1^; NH_4_OAc-exchangeable K was 89.4 ± 1.1 mg kg^−1^; and the pH was 6.45 ± 0.07. In mid-April of 2016 and 2017, rice seeds were sown in the seedbed and transplanted in mid-May with a hill spacing of 30 cm × 10 cm, with three plants per hill. The fertilization standard was composed of nitrogen (150 kg per ha as urea), phosphorus (100 kg per ha as diammonium phosphate) and potassium (75 kg per ha as potassium sulfate). Urea was also used at mid-tillering (100 kg per ha) as a top dressing.

### 4.2. Experimental Design

A split-plot design was used for this experiment. Drought stress with different soil water at the jointing–booting stage was set as the whole-plot factor with three levels, mild drought stress (−10 kPa), moderate drought stress (−25 kPa) and severe drought stress (−40 kPa); irrigation with 3 cm depth was used as a control (0 kPa). The water potential of each treatment of SJ6 was marked as A0 (0 kPa), A1 (−10 kPa), A2 (−25 kPa) and A3 (−40 kPa) and DN425 was marked as B0 (0 kPa), B1 (−10 kPa), B2 (−25 kPa) and B3 (−40 kPa). The booting stage was determined by manually dissecting the stem and visually observing it. The subplot factor was rice variety and the area of the subplot was 35 m^2^ (4 m × 9 m, approximately). A large soil ridge was used to separate the plots (Appendix A, 60 cm width). The booting stage was defined as 50% of the plants having the panicle that was visible to the naked eye as a tiny and transparent growth < 2 mm in length buried within the leaf sheaths near the base of the plant, approximately two weeks before heading. The drought stress treatment lasted for 15 d. The zero point is recorded before the beginning of water stress, marked D0 days. The days after each treatment that reach the design water potential are recorded as the days after stress. Irrigation was controlled before the drought stress to gradually reduce the soil water potential. A small amount of water was added to maintain the potential for the design if there was a water shortage during the comparable stress period. All treatments were placed under a rain-proof shelter during the drought treatment period. Eight soil tensiometers were placed evenly in two rows in each plot (Appendix A). A soil tensiometer (Institute of Soil Science, Chinese Academy of Sciences, Nanjing, China) was used to monitor the soil water potential at a depth of 20 cm. The soil potential was monitored daily at 6:00, 12:00 and 18:00. When the soil water potential was low, an appropriate water supply was used to maintain the soil water potential within a range of ±3 kPa of the designed water potential. Regular irrigation was restored immediately after the drought stress. For the convenience of description, days after drought stress were recorded as Dx days and days after restoration of irrigation were recorded as Rx days.

### 4.3. Determination of Indexes

#### 4.3.1. Sample Collection

The sampling time of the two-year parallel test is shown in Figure 1. Panicles were collected from each treatment (they were not sampled before D6 days because of their small size). Three grains at the top of the upper three primary branches were the superior grains and three grains at the end of the lower three secondary branches were the inferior grains. Superior and inferior grains were selected and stored in liquid nitrogen to determine the starch synthesis-related enzymes present in them.

#### 4.3.2. Amylose and Amylopectin

Starch content in grains was determined using the dual-wavelength method. The primary wavelength of amylopectin visibility was 550 nm and the reference wavelength was 730 nm. The primary wavelength of amylose visibility was 620 nm and the reference wavelength was 430 nm.

#### 4.3.3. Soluble Starch Synthase and Granule Bound-Starch Synthase (GBSS)

The activity of soluble starch synthase and GBSS was determined as previously described [42], with some modifications. Thus, 5–10 grains were ground with a mortar and pestle in liquid nitrogen and homogenized in 1 mL extracting solution (50 mM HEPES-NaOH (pH 7.5), 50 mM MgCl_2_, 10 mM DTT, 2% PVP). The homogenate was centrifuged at 10,000× *g* for 10 min at 4 °C. The supernatant was soluble starch synthase crude enzyme solution and the precipitate was added to 1 mL extract for suspension. The turbid liquid was a GBSS crude enzyme solution. Amylopectin (0.7 mg) was added to a 75 μL crude enzyme solution in a 30 °C water bath for 20 min and boiled in a water bath for 1 min. Then, 75 μL working solution (200 mM KCl, 10 mM MgCl_2_, 4 mM PEP, 1.2 U pyruvate kinase) was added to the mixture, followed by a 30 °C water bath for 30 min and then boiled in a water bath for 1 min. Rapid ice bath and centrifugation at 4 °C for 10 min at 10,000 rpm. Following this, 150 μL supernatant was mixed with 100 μL reaction solution (10 mM glucose, 20 mM MgCl_2_, 2 mM NADP), 5 μL 1.4 U hexokinase and 5 μL 0.35 U glucose 6-phosphate dehydrogenase and the absorbance was measured at 340 nm. The catalyzed production of 1 nmol NADPH per minute per grain is defined as one unit of enzyme activity.

#### 4.3.4. Starch Branching Enzyme (SBE)

The starch branching enzyme activity was determined as described by referring to [42]. Five to ten grains were ground with a mortar and pestle in liquid nitrogen and homogenized in 1 mL extracting solution (50 mM HEPES-NaOH (pH 7.5), 50 mM MgCl_2_, 10 mM DTT, 2% PVP). The homogenate was centrifuged at 10,000× *g* for 10 min at 4 °C. The supernatant was a starch-branching enzyme–crude enzyme solution. Eighty-five microliters of HEPES-NaOH and 1 mg soluble starch were added to 65 μL crude enzyme solution and inactivated crude enzyme solution, respectively. Then, water baths were run at 37 °C for 20 min and 95 °C for 5 min. One hundred and thirty microliters of 1 M HCl and 10 μL of I_2_-KI (0.1% I_2_ and 1% KI) were added successively and homogenized at room temperature (25 °C) for 10 min and the absorbance was measured at 660 nm. The blue iodine value of each grain decreased by 1%, defined as 1 unit of enzyme activity.

#### 4.3.5. Dry Matter Accumulation and Grain Filling Dynamics

Three hills of representative plants were selected for each treatment every three days after drought stress and every five days after irrigation restoration. The plants were separated according to organs, such as the stem and sheath, leaf and panicle. The plants were killed by subjecting them to 105 °C for 30 min and subsequently drying to a constant weight at 80 °C. The dry matter accumulation was measured and grain-filling parameters including the mean grain-filling rate (Gmean), the maximum grain-filling rate (Gmax), time to reach the maximum grain-filling rate (Tmax) and time of grain-filling active period (D) were calculated using the Richards equation according to Zhu [43].


$$\mathrm{Richards}\;\mathrm{equation}:\;W=\frac{A}{{\left(1+{\mathrm{Be}}^{-\mathrm{Kt}}\right)}^{\frac{1}{N}}};(\mathrm{GRmean})=\frac{\mathrm{AK}}{\left[2(N+2)\right]};(\mathrm{GRmax})=\frac{\mathrm{AK}}{{\left(1+N\right)}^{\frac{\left(N+1\right)}{N}}};\;(\mathrm{Tmax})=\frac{\left(\mathrm{lnB}-\mathrm{lnN}\right)}{K};(D)=\frac{\left[2(N+2)\right]}{K}$$


### 4.4. Yield and Yield Components

After complete ripening, 15 hills of representative plants were selected for plant laboratory testing, followed by determination of panicle traits and yield components. Grain yield was obtained from all plants in a 1 m^2^ area (except border plants) in each plot and adjusted to a moisture content in 0.14 g H_2_O g^−1^ fresh weight.

### 4.5. Statistical Analysis

Data analyses were performed using the SPSS 18.0 (Chicago, IL, USA) software package. Analysis of variance (ANOVA) was used to analyze all data and differences among treatments. Results are reported as the mean ± standard deviation (SD) values of three independent experiments, measuring at least three different replicates (plants) in each experiment. SD was calculated directly from crude data. The levels of significance in the figures are given by ns for not significant, * and ** for significant at *p* < 0.05 and *p* < 0.01, respectively.

## 5. Conclusions

Dry matter accumulation was significantly reduced in organs under drought stress at the jointing–booting stage and eventually decreased the grain number per panicle and thousand-grain weight. Severe drought inhibits dry matter transport and a large amount of dry matter remains in vegetative organs. The high content of NSCs in stem and sheath plays a key role in resisting drought stress. Under severe and severe drought conditions, two different types of rice had different NSC metabolic strategies. The decrease in dry matter translocation after anthesis directly caused a change in the grain-filling strategy. The grain-filling rate decreased significantly with an increase in the drought stress. The active grain-filling time of the superior grains was shortened to complete the necessary reproductive growth. In comparison, the filling time of inferior grains was significantly prolonged to avoid a dramatic reduction in yield. The significant decrease in the grain-filling rate of the superior and inferior grains caused a reduction in the thousand-grain weight. In particular, the influence of the grain-filling rate of inferior grains on the thousand-grain weight was more significant. Drought stress changed the starch synthesis strategy of superior and inferior grains. Amylose content was decreased and amylopectin synthesis was enhanced under drought stress, especially in inferior grains.

## Figures and Tables

**Figure 1 ijms-23-11157-f001:**
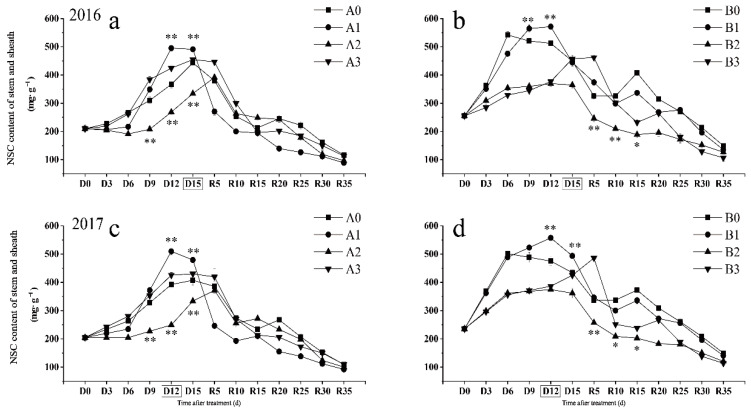
Effects of drought stress at the jointing–booting stage on NSC content in growth period overlapping rice in 2016 (**a**,**b**) and 2017 (**c**,**d**). The full heading stage was D15 days in 2016 and D12 days in 2017. Vertical bars represent standard deviation. A0 and B0, control treatments (0 kPa) of SJ6 and DN425; A1 and B1, mild drought stress treatments (-10 kPa); A2 and B2, mild drought stress treatments (-25 kPa); A3 and B3, severe drought stress (-40 kPa). *, ** Represent significance at *p* < 0.05 and *p* < 0.01, respectively.

**Figure 2 ijms-23-11157-f002:**
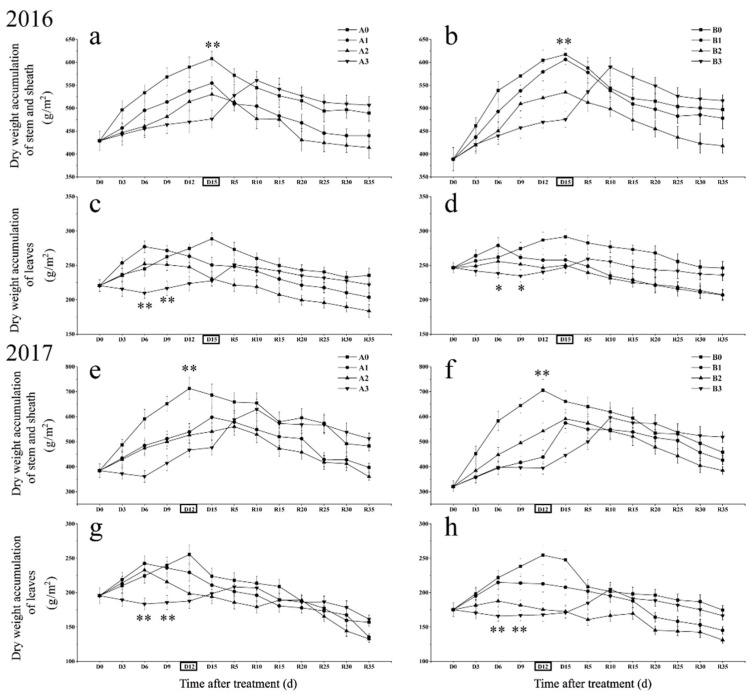
Effects of drought stress at the jointing–booting stage on dry matter accumulation in growth period overlapping rice in 2016 (**a**–**d**) and 2017 (**e**–**h**). The full heading stage was D15 days in 2016 and D12 days in 2017. Vertical bars represent standard deviation. A0 and B0, control treatments (0 kPa) of SJ6 and DN425; A1 and B1, mild drought stress treatments (−10 kPa); A2 and B2, mild drought stress treatments (−25 kPa); A3 and B3, severe drought stress (−40 kPa). ** Represent significance at *p* < 0.01, respectively.

**Figure 3 ijms-23-11157-f003:**
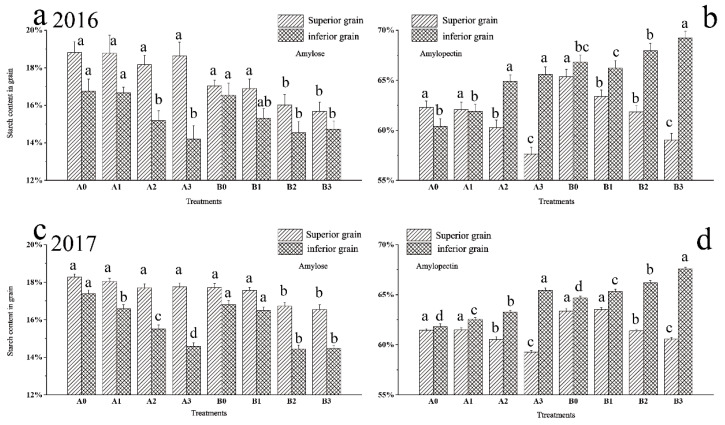
Effects of drought stress at the jointing–booting stage on starch components in superior and inferior grains in growth period overlapping rice in 2016 (**a**,**b**) and 2017 (**c**,**d**). Vertical bars represent standard deviation. Values for the same grain type and the same varieties followed by different letters are significantly different at *p* = 0.05. A0 and B0, control treatments (0 kPa) of SJ6 and DN425; A1 and B1, mild drought stress treatments (−10 kPa); A2 and B2, mild drought stress treatments (−25 kPa); A3 and B3, severe drought stress (−40 kPa).

**Figure 4 ijms-23-11157-f004:**
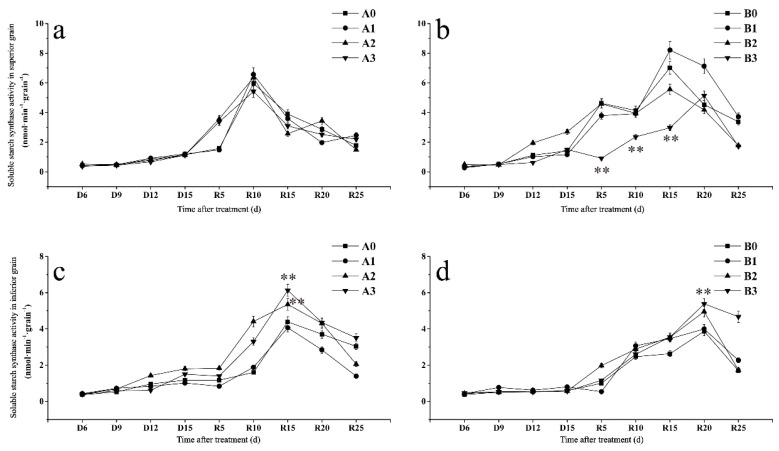
Effects of drought stress at the jointing–booting stage on soluble starch synthase activity of superior (**a**,**b**) and inferior (**c**,**d**) grains in growth period overlapping rice. ** Represent significance at *p* < 0.01, respectively.

**Figure 5 ijms-23-11157-f005:**
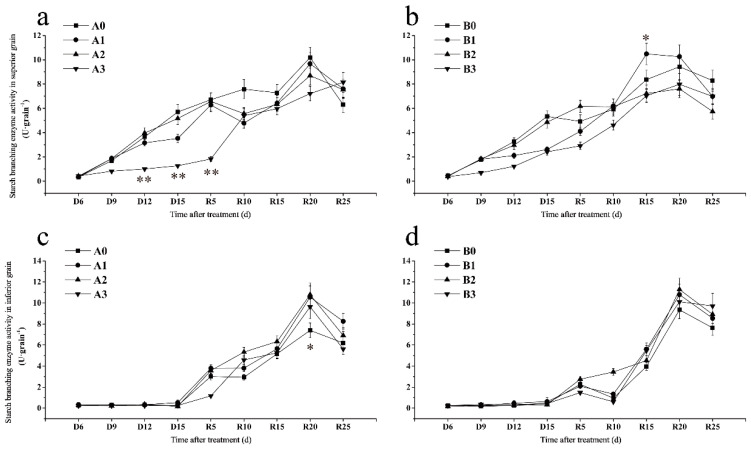
Effects of drought stress at the jointing–booting stage on the key enzymes for starch synthesis of superior (**a**,**b**) and inferior (**c**,**d**) grains in growth period overlapping rice. *, ** Represent significance at *p* < 0.05 and *p* < 0.01, respectively.

**Figure 6 ijms-23-11157-f006:**
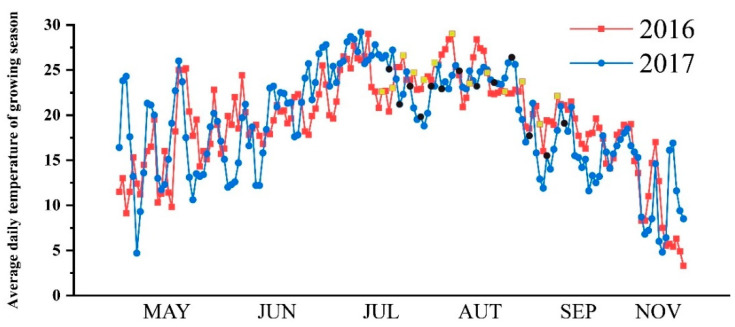
Average daily temperature (°C) of growing season in experimental site. The sampling dates were marked yellow squares (2016) and black circles (2017).

**Table 1 ijms-23-11157-t001:** Effects of drought stress at the jointing–booting stage on dry matter translocation in growth period overlapping rice in 2016 and 2017.

Year	Treatment	DMT		DMTE		DMTCRV	
		StSh	Leaf	StSh	Leaf	StSh	Leaf
2016	A0	118.741 ^a^	53.127 ^a^	19.526% ^b^	18.406% ^b^	16.911% ^b^	7.566% ^b^
	A1	114.643 ^a^	46.795 ^b^	20.671% ^a^	18.684% ^b^	24.321% ^a^	9.927% ^a^
	A2	116.126 ^a^	46.954 ^b^	21.901% ^a^	20.361% ^a^	25.852% ^a^	10.453% ^a^
	A3	20.965 ^b^	28.836 ^c^	3.973% ^c^	11.497% ^c^	6.037% ^c^	8.304% ^b^
	B0	120.110 ^a,b^	45.391 ^b^	19.460% ^b^	15.568% ^c^	19.610% ^b^	7.411% ^b^
	B1	128.089 ^a^	50.888 ^a^	21.127% ^a^	19.734% ^a^	25.001% ^a^	9.932% ^a^
	B2	116.788 ^b^	42.980 ^b^	21.834% ^a^	17.176% ^b^	28.607% ^a^	10.528% ^a^
	B3	19.324 ^c^	23.503 ^c^	3.605% ^c^	9.059% ^d^	5.526% ^c^	6.721% ^b^
2017	A0	203.680 ^a^	67.108 ^a,b^	29.672% ^b^	29.962% ^c^	23.468% ^b^	7.732% ^b^
	A1	200.401 ^a^	75.450 ^a^	33.538% ^a^	35.796% ^a^	30.591% ^a^	11.518% ^a^
	A2	180.365 ^a^	61.554 ^b^	33.366% ^a^	31.692% ^b^	29.486% ^a^	10.063% ^a^
	A3	−36.175 ^b^	37.105 ^c^	−7.592% ^c^	18.688% ^d^	−8.702% ^c^	8.926% ^b^
	B0	203.026 ^a^	73.076 ^a^	30.726% ^b^	29.497% ^a^	24.522% ^b^	8.826% ^a^
	B1	148.879 ^b^	62.720 ^b^	25.897% ^c^	30.176% ^a^	22.379% ^b^	9.428% ^a^
	B2	206.152 ^a^	41.404 ^c^	34.856% ^a^	23.988% ^b^	34.881% ^a^	7.006% ^b^
	B3	−72.853 ^c^	4.395 ^d^	−16.347% ^d^	2.571% ^c^	−15.847% ^c^	0.956% ^c^

Values for the same organ and the same varieties followed by different letters are significantly different at *p* = 0.05. DMT-dry matter translocation; DMTE-dry matter translocation efficiency; DMTCRV-dry matter translocation conversion rate of the vegetative organ. StSh, stem and sheath. A0 and B0, control treatments (0 kPa) of SJ6 and DN425; A1 and B1, mild drought stress treatments (−10 kPa); A2 and B2, mild drought stress treatments (−25 kPa); A3 and B3, severe drought stress (−40 kPa).

**Table 2 ijms-23-11157-t002:** Effects of drought stress at the jointing–booting stage on grain-filling characteristics of superior and inferior grains in growth period overlapping rice in 2016 and 2017.

Year	Treatment	GRmean		GRmax		Tmax (d)		D (d)	
		SG	IG	SG	IG	SG	IG	SG	IG
2016	A0	0.875 ^a^	0.573 ^a^	1.363 ^a^	0.938 ^a^	12.937 ^c^	20.460 ^b^	26.193 ^b^	30.652 ^c^
	A1	0.696 ^c^	0.524 ^b^	1.114 ^b^	0.837 ^b^	14.910 ^b^	25.077 ^a^	31.229 ^a^	34.103 ^b^
	A2	0.729 ^b^	0.498 ^c^	1.095 ^b^	0.766 ^b^	11.167 ^d^	20.416 ^b^	31.045 ^a^	39.951 ^a^
	A3	0.786 ^b^	0.418 ^d^	1.394 ^a^	0.664 ^c^	22.175 ^a^	25.073 ^a^	23.378 ^c^	41.784 ^a^
	B0	0.949 ^a^	0.661 ^a^	1.472 ^a^	1.067 ^a^	16.707 ^b^	25.003 ^b^	26.878 ^c^	31.389 ^c^
	B1	0.816 ^c^	0.633 ^a^	1.245 ^b^	1.031 ^a^	14.560 ^d^	26.290 ^a^	30.445 ^a^	32.074 ^c^
	B2	0.799 ^d^	0.526 ^b^	1.240 ^b^	0.814 ^b^	15.301 ^c^	23.041 ^d^	28.695 ^b^	41.620 ^b^
	B3	0.844 ^b^	0.461 ^c^	1.396 ^a^	0.698 ^c^	22.387 ^a^	24.337 ^c^	25.163 ^d^	47.791 ^a^
2017	A0	0.882 ^a^	0.570 ^a^	1.392 ^a^	0.933 ^a^	16.334 ^c^	23.602 ^b^	25.754 ^c^	30.946 ^b^
	A1	0.691 ^c^	0.533 ^a,b^	1.116 ^c^	0.861 ^a,b^	16.256 ^c^	26.103 ^a^	31.370 ^a^	32.840 ^b^
	A2	0.801 ^b^	0.502 ^b^	1.237 ^b^	0.777 ^b^	16.641 ^b^	24.551 ^a,b^	27.931 ^b^	38.864 ^a^
	A3	0.790 ^b^	0.439 ^c^	1.386 ^a^	0.701 ^b^	20.908 ^a^	23.893 ^a,b^	23.301 ^d^	39.300 ^a^
	B0	0.929 ^a^	0.676 ^a^	1.439 ^a^	1.100 ^a^	18.425 ^c^	27.221 ^a^	26.902 ^c^	30.539 ^d^
	B1	0.815 ^c^	0.621 ^b^	1.251 ^b^	1.007 ^b^	15.856 ^d^	27.129 ^a,b^	30.374 ^a^	32.639 ^c^
	B2	0.801 ^d^	0.522 ^c^	1.253 ^b^	0.812 ^c^	19.616 ^b^	27.015 ^b^	28.456 ^b^	41.206 ^b^
	B3	0.846 ^b^	0.458 ^d^	1.393 ^a^	0.700 ^d^	20.271 ^a^	23.281 ^c^	25.187 ^d^	46.443 ^a^

Values for the same grain type and the same varieties followed by different letters are significantly different at *p* = 0.05. GRmean, mean grain-filling rate; GRmax, maximum grain-filling rate; Tmax, time of reaching the maximum grain-filling rate; D, grain-filling active period. SG, superior grains; IG, inferior grains. A0 and B0, control treatments (0 kPa) of SJ6 and DN425; A1 and B1, mild drought stress treatments (−10 kPa); A2 and B2, mild drought stress treatments (−25 kPa); A3 and B3, severe drought stress (−40 kPa).

**Table 3 ijms-23-11157-t003:** Effects of drought stress at the jointing–booting stage on the panicle characters of growth period overlapping rice in 2016 and 2017.

Variety (V)	Treatment (T)	Panicle Length (cm)		PBN		GNPB		SBN		GNSB	
		2016	2017	2016	2017	2016	2017	2016	2017	2016	2017
SJ6	A0	14.862 ^a^	15.054 ^a^	9.122 ^a^	9.936 ^a^	48.020 ^a^	53.562 ^a^	12.622 ^a^	12.787 ^a^	34.127 ^a^	38.139 ^a^
	A1	13.561 ^a^	14.199 ^a,b^	8.499 ^a,b^	8.744 ^b^	44.129 ^a^	47.345 ^a^	9.836 ^b^	9.614 ^b^	25.350 ^b^	31.887 ^b^
	A2	11.511 ^b^	13.425 ^b^	8.414 ^a,b^	9.111 ^b^	43.321 ^a^	49.065 ^a^	8.095 ^c^	8.019 ^c^	20.592 ^c^	26.870 ^c^
	A3	9.147 ^c^	11.954 ^c^	7.532 ^b^	8.236 ^b^	38.742 ^a^	45.774 ^a^	5.496 ^d^	6.412 ^d^	14.334 ^d^	18.440 ^d^
DN425	B0	16.583 ^a^	17.679 ^a^	8.847 ^a^	8.263 ^a^	42.195 ^a^	43.877 ^a^	11.559 ^a^	13.297 ^a^	30.573 ^a^	36.204 ^a^
	B1	15.095 ^b^	16.206 ^a,b^	7.513 ^b^	7.706 ^a^	37.462 ^a,b^	42.236 ^a^	8.795 ^b^	10.458 ^b^	22.846 ^b^	32.499 ^a^
	B2	13.971 ^b^	15.015 ^b,c^	7.537 ^b^	7.841 ^a^	36.216 ^a,b^	44.101 ^a^	7.723 ^b^	8.043 ^c^	19.716 ^b^	23.354 ^b^
	B3	12.309 ^c^	14.089 ^c^	6.795 ^b^	8.296 ^a^	31.327 ^b^	44.271 ^a^	5.665 ^c^	6.858 ^c^	12.193 ^c^	17.358 ^c^
Source	df	Mean square									
V	1	55.691 **		8.655 *		436.912 **		0.044 ns		42.164 **	
T	3	38.172 **		3.720 **		96.045 **		90.301 **		796.655 **	
Y	1	20.998 **		2.816 **		446.874 **		6.086 **		380.025 **	
V × T	3	0.409 ns		0.353 ns		5.463 ns		0.196 ns		1.791 ns	
V × Y	1	0.051 ns		0.206 ns		6.200 ns		3.197 *		1.861 ns	
T × Y	3	1.630 ns		0.589 ns		26.243 ns		0.523 ns		8.172 ns	
V × T × Y	3	0.696 ns		0.618 ns		12.142 ns		0.498 ns		4.486 ns	
Error	32	0.603		0.256		9.976		0.628		5.459	

Values for the same year and the same varieties followed by different letters are significantly different at *p* = 0.05. PBN: Primary branch number, GNPB: Grain number of primary branch, SBN: Secondary branch number, GNSB: Grain number of secondary branch. A0 and B0, control treatments (0 kPa) of SJ6 and DN425; A1 and B1, mild drought stress treatments (−10 kPa); A2 and B2, mild drought stress treatments (−25 kPa); A3 and B3, severe drought stress (−40 kPa). *, ** represent significance at *p* < 0.05 and *p* < 0.01, respectively. And ns represent non-significant.

**Table 4 ijms-23-11157-t004:** Effects of drought stress at the jointing–booting stage on the yield and its composition of growth period overlapping rice in 2016 and 2017.

Variety (V)	Treatment (T)	EPN (Panicel·m^−2^)		SPP		Seed Setting Rate		TGW (g)		Theoretical Yield (kg·hm^−2^)		Actual Yield (kg·hm^−2^)	
		2016	2017	2016	2017	2016	2017	2016	2017	2016	2017	2016	2017
SJ6	A0	478.5 ^a^	473.0 ^a^	82.147 ^a^	91.700 ^a^	96.03% ^a,b^	93.25% ^a^	24.246 ^a^	24.670 ^a^	9143.151 ^a^	9951.101 ^a^	7021.7 ^a^	8678.9 ^a^
	A1	401.5 ^a^	451.0 ^a^	69.478 ^b^	79.233 ^b^	97.29% ^a^	92.78% ^a^	23.840 ^a,b^	23.620 ^b^	6499.060 ^b^	7825.292 ^b^	4713.8 ^b^	6550.9 ^b^
	A2	423.5 ^a^	462.0 ^a^	63.912 ^b^	75.935 ^b^	96.04% ^a,b^	94.61% ^a^	23.349 ^a,b^	23.580 ^b^	6046.034 ^b^	7815.037 ^b^	4491.9 ^b^	6117.1 ^c^
	A3	412.5 ^a^	440.0 ^a^	53.076 ^c^	64.214 ^c^	94.77% ^b^	93.56% ^a^	22.728 ^b^	23.080 ^b^	4706.740 ^c^	6083.565 ^c^	3472.5 ^c^	4157.1 ^d^
DN425	B0	473.0 ^a^	495.0 ^a^	72.768 ^a^	80.081 ^a^	96.42% ^a^	94.58% ^a^	26.911 ^a^	26.360 ^a^	8925.619 ^a^	9949.520 ^a^	6125.1 ^a^	8279.4 ^a^
	B1	440.0 ^a^	473.0 ^a^	60.308 ^b^	74.735 ^a,b^	95.07% ^a^	90.86% ^a^	25.385 ^b^	25.370 ^b^	6410.777 ^b^	8107.957 ^b^	5123.4 ^b^	6652.7 ^b^
	B2	434.5 ^a^	440.0 ^a^	55.933 ^b^	67.455 ^b,c^	95.85% ^a^	92.02% ^a^	25.028 ^b,c^	25.160 ^b,c^	5828.906 ^c^	6851.691 ^c^	4682.5 ^c^	5910.1 ^c^
	B3	429.0 ^a^	429.0 ^a^	43.520 ^c^	61.628 ^c^	97.32% ^a^	92.34% ^a^	24.264 ^c^	24.460 ^c^	4402.667 ^d^	5963.277 ^d^	3496.7 ^d^	4597.2 ^d^
Source	df	Mean square											
V	1	958.547 ns		750.532 **		0.000 ns		35.837 **		4.98 × 10^5^ ns			
T	3	6146.422 ns		1411.595 **		0.000 ns		7.596 **		3.68 × 10^7^ **			
Y	1	5450.672 ns		1651.093 **		0.120 **		0.570 ns		2.10 × 10^7^ **			
V × T	3	701.422 ns		11.346 ns		0.001 *		0.291 ns		2.49 × 10^5^ ns			
V × Y	1	459.422 ns		14.854 ns		0.000 ns		0.197 ns		112.218 ns			
T × Y	3	625.797 ns		19.369 ns		0.000 ns		0.107 ns		2.28 × 10^5^ ns			
V × T × Y	3	565.297 ns		13.967 ns		0.000 ns		0.191 ns		1.94 × 10^5^ ns			
Error	32	3.80 × 10^3^		15.858		0.000		0.234		1.33 × 10^6^			

Values for the same year and the same varieties followed by different letters are significantly different at *p* = 0.05. EPN: Efficient panicle number, SPP: Spikelets per panicle, TGW: Thousand-grain weight. A0 and B0, control treatments (0 kPa) of SJ6 and DN425; A1 and B1, mild drought stress treatments (−10 kPa); A2 and B2, mild drought stress treatments (−25 kPa); A3 and B3, severe drought stress (−40 kPa). *, ** represent significance at *p* < 0.05 and *p* < 0.01, respectively. And ns represent no significance.

**Table 5 ijms-23-11157-t005:** Correlation of grain-filling characteristics with yield components.

	Yield	DMT	EPN	SPP	TGW	SGRmean	IGRmean	SGD	IGD
Yield	—								
DMT	0.806 **	—							
EPN	0.866 **	0.682 **	—						
SPP	0.906 **	0.691 **	0.782 **	—					
TGW	0.529 *	0.403	0.514 *	0.179	—				
SGRmean	0.487	0.134	0.643 **	0.303	0.692 **	—			
IGRmean	0.815 **	0.659 **	0.733 **	0.574 *	0.785 **	0.522 *	—		
SGD	0.279	0.462	−0.049	0.182	0.174	−0.434	0.421	—	
IGD	−0.859 **	−0.638 **	−0.798 **	−0.815 **	−0.438	−0.446	−0.847 **	−0.235	—

DMT, dry matter translocation; EPN, efficient panicle number; SPP, spikelet per panicle; TGW, thousand-grain weight; SGRmean, average grain-filling rate of superior grain; IGDmean, average grain-filling rate of inferior grain. SGD, Filling active period of superior grain; IGD, Filling active period of inferior grain; * represents significance at *p* < 0.05; ** represents significance at *p* < 0.01.

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
