# Peer review of "Response of Rice with Overlapping Growth Stages to Water Stress by Assimilates Accumulation and Transport and Starch Synthesis of Superior and Inferior Grains"

_ijms, 2022, doi:10.3390/ijms231911157_

Round 1

Reviewer 1 Report

The authors investigated the dry matter translocation and starch synthesis in superior and inferior grains of rice under different drought stress. They did a valuable experiment and measured good traits. They showed that drought stress affected yield components and NSC content in stem increase the resistance to drought stress.

Authors used a split-plot design (Line 611and Fig. S1) to do the experiment but analyzed the data based on a factorial design (tables 3 & 4). They repeated the experiment in 2016 and 2017. The data should be re-analyzed based on a combined split plot design, and SPSS doesn’t have the capability to do this analysis.

Line 618. Give a measurement (how many meters?) to show the separation between main plot and subplot.

Reviewer 2 Report

The work by Wang et al describes the effect of drought on rice at the overlapping stages of jointing and booting. Specifically they analyse the effect of mild, moderate and severe drought stress on dry matter accumulation and translocation. They show a contrast in superior and inferior grain filling with the first supporting reproductive growth and the second grain yield.

Although the results of the study are somehow expected, the authors provide novel evidence which is supported by adequate statistical analysis.

It would be advisable to provide images of the growth stages they decribe of the respective tissues in the supplementary data.
